# Early attentional processing and cortical remapping strategies of tactile stimuli in adults with an early and late-onset visual impairment: A cross-sectional study

Mónica-Alba Ahulló-Fuster[1], M. Luz Sánchez-Sánchez[2]*, Enrique Varela-Donoso[1], Tomás Ortiz[3]

1 Department of Radiology, Rehabilitation and Physiotherapy, Faculty of Nursing, Physiotherapy and Podiatry, Complutense University of Madrid, Madrid, Spain, 2 Physiotherapy in Motion, Multispeciality Research Group (PTinMOTION), Department of Physiotherapy, University of Valencia, Valencia, Spain, 3 Department of Legal Medicine, Psychiatry and Pathology, Faculty of Medicine, Complutense University of Madrid, Madrid, Spain

* M.Luz.Sanchez@uv.es

**Data Availability Statement:** The data are held in a public repository: https://doi.org/10.7910/DVN/K7IGAB.

## Abstract

Neuroplastic changes appear in people with visual impairment (VI) and they show greater tactile abilities. Improvements in performance could be associated with the development of enhanced early attentional processes based on neuroplasticity. Currently, the various early attentional and cortical remapping strategies that are utilized by people with early (EB) and late-onset blindness (LB) remain unclear. Thus, more research is required to develop effective rehabilitation programs and substitution devices. Our objective was to explore the differences in spatial tactile brain processing in adults with EB, LB and a sighted control group (CG). In this cross-sectional study 27 participants with VI were categorized into EB (n = 14) and LB (n = 13) groups. They were then compared with a CG (n = 15). A vibrotactile device and event-related potentials (ERPs) were utilized while participants performed a spatial tactile line recognition task. The P100 latency and cortical areas of maximal activity were analyzed during the task. The three groups had no statistical differences in P100 latency (p>0.05). All subjects showed significant activation in the right superior frontal areas. Only individuals with VI activated the left superior frontal regions. In EB subjects, a higher activation was found in the mid-frontal and occipital areas. A higher activation of the mid-frontal, anterior cingulate cortex and orbitofrontal zones was observed in LB participants. Compared to the CG, LB individuals showed greater activity in the left orbitofrontal zone, while EB exhibited greater activity in the right superior parietal cortex. The EB had greater activity in the left orbitofrontal region compared to the LB. People with VI may not have faster early attentional processing. EB subjects activate the occipital lobe and right superior parietal cortex during tactile stimulation because of an early lack of visual stimuli and a multimodal information processing. In individuals with LB and EB the orbitofrontal area is activated, suggesting greater emotional processing.

**Funding:** This work was supported by a Complutense University of Madrid's pre-doctoral fellowship for research personnel in training (CT63/19-CT64/19), awarded to MAAF. The funders had no role in study design, data collection and analysis, decision to publish or preparation of the manuscript.

**Competing interests:** The authors have declared that no competing interests exist.

## Introduction

Neuroplasticity is a property of the human nervous system by which neurons change their connectivity in various forms [1]. It occurs while learning or due to sensorial and cognitive stimulation [2]. Plastic changes help us understand the typical cognitive functions and the compensatory modifications that occur after an injury, such as brain damage or blindness [3–5].

Visual impairment (VI) exists worldwide. Approximately 295 million people have a moderate to severe visual deficit, and 43.3 million have total blindness. Moreover, the prevalence of VI increases with an aging global population [6]. VI may have several origins, such as glaucoma, retinitis pigmentosa, and optic nerve atrophy. Undeniably, the loss of vision has a detrimental effect on an individual's quality of life, mobility, and independence [7]. This demonstrates the need for a rehabilitation program to boost the functionality and independence of those with a VI [8]. For that reason, several researchers have developed tactile substitution systems for VI subjects [9, 10].

Tactile information helps supply data deficits resulting from faulty visual and auditive channels [11]. Furthermore, neuroimaging studies have shown that tactile recognition of objects activates the occipital visual cortex in people with VI, making them more skilled than their sighted counterparts [12–15]. Initially, the lateral occipital cortex (LOC) was supposed to be an area specialized in the visual recognition of objects. However, it has also been proven to be active during tactile recognition [16, 17]. Therefore, this region is a clear example of multimodal spatial information processing [2]. These findings reinforce the hypothesis of substitutive neuroplasticity in people with VI [13].

It is suggested that neuroplasticity could be behind the superior tactile skills found in people with VI. Some authors have hypothesized that a faster early attentional processing mechanism based on a neuroplastic compensatory system enables people with blindness to recognize spatial stimuli better [11, 17–19]. In contrast, others claim that the superior tactile abilities of people with VI are a consequence of cognitive enhancement, rather than faster early attentional processing [20–22]. Furthermore, none of the previous studies have explored the variations in early attentional processing depending on the age onset of blindness.

Somatosensory event-related potentials (ERPs) can reveal valuable data about processing spatial tactile information in the cortex [23]. After a stimulus is presented, the electroencephalogram typically exhibits a pattern of oscillations, including the P100 waveform, nestled amidst a series of peaks and troughs. The emergence of P100 activity is believed to occur in primary sensorial cortical areas, offering insight into the initial phases of the neuronal processing within the neocortical circuit [24–27]. There is growing consensus concerning the contribution of P100 waveform in the context of global cognitive processing, notably in situations involving attention. In paradigms in which attention must be directed to a stimulus in space, the ERP P100 has been observed to be modulated by early attentional activity [24]. Spatial attention modulates neural activity as early as 100 ms after stimulus presentation [28, 29]. Top-down modulation of cortical activity during the early phases of perceptual processing influences subsequent working-memory management [30]. Thus, neuroplastic changes in VI and early attentional processing can be explored using tools such as ERPs along with quantitative electroencephalography [17].

To get insight into early attentional processing in people with VI, in this study, a passive vibrotactile device was used to compare the spatial early attentional brain processing between sighted individuals and adults with early and late-onset VI, when carrying out a simple spatial tactile task. We also set out to analyze the maximum brain activity in each group and to compare the brain activity in subjects with early and late-onset VI and a group of sighted

individuals. It was hypothesized that individuals with a VI might show different early attentional processing and brain activity depending on the age of VI acquisition, and different from sighted people.

## Materials and methods

### Participants

A convenience sample of 42 adults (27 with a VI and 15 controls) were recruited during the second trimester of 2020 for this cross-sectional study. Participants were enlisted from associations for people with VI and universities in Madrid. The selection criteria included being between 18 and 60 years old, having a type of VI that implies total blindness or a severe visual deficit according to the International Classification of Diseases 11[th] revision [31], having a loss of vision before five years or after 14 years of age, and being capable of understanding simple verbal instructions. Participants with vestibular, neuropsychiatric, brain damage, or sensorimotor disorders, were excluded. The criteria for having a VI were verified through a medical report provided by the participant [32]. Thus, participants with a VI were categorized into two groups: 1) early-onset blindness (EB) (loss of vision before five years of age) and 2) late-onset blindness (LB) (loss of vision after 14 years of age) in accordance with previous research [33, 34]; and based on the developmental stages of the anatomical region of V1 described by Siu and Murphy [35].

Age was used as a criterion to match individuals of the control group (CG), people with no history of visual, sensorimotor, vestibular, or neuropsychiatric disorders. They were recruited from family members of the participants with VI and the authors' institution.

All participants were informed about the study procedures verbally and in writing. They then provided informed written consent for their participation. In order to protect the identity of the participants, each individual was associated with a numerical code. This information was only available to the principal investigator. This study was in full compliance with the Declaration of Helsinki and was approved by the San Carlos University Hospital Ethical Committee (20/071-E_Tesis). It is important to note that this study was conducted per the STrengthening the Reporting of OBservational studies in Epidemiology guidelines [36].

### Procedures and lines orientation task

Data collection was performed at the Department of Legal Medicine, Psychiatry and Pathology of the Faculty of Medicine of the Complutense University of Madrid during the second and third trimester of 2021. The demographical and clinical data of the subjects were recorded during an interview. After that, the participants were required to remain as calm as possible while sitting on an armchair in front of a table where the tactile stimulator device was placed. The room used for the experiment was completely dark and isolated from external noise to ensure equal conditions for all the participants. Then, the participants placed their left hand on the stimulator device with their right hand on a keyboard. The tactile stimulator (stimulation matrix) was positioned against the palm of the volunteer, and the spatial tactile stimuli were passively received into the palm (Fig 1). The stimulation matrix had 28×28 stimulation points aimed at showing the vibrotactile stimuli. The stimuli were projected in the stimulation matrix at a rate of one stimulus per second [17].

Three hundred lines of stimuli were presented in the stimulator matrix. Following the oddball paradigm, 80% of the lines were oriented vertically and 20% horizontally. The sequence of line presentation was randomized, and the total time required to complete the task was five minutes. The horizontal line (the low-frequency stimuli) was designed as the target for the

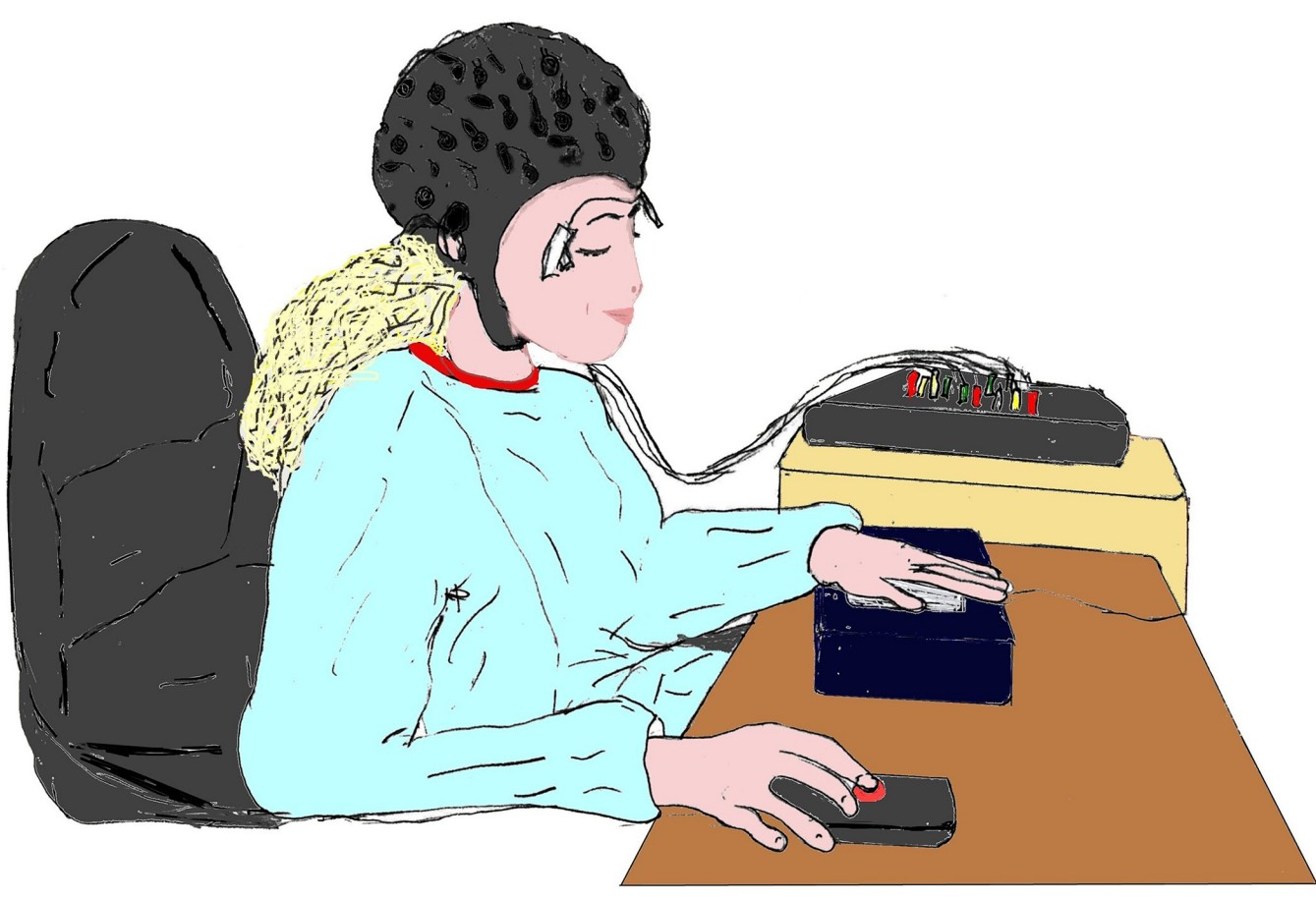

**Fig 1. Schematic representation of the passive stimulation task.**

ERPs. The participants were asked to actively press the keyboard (motor response) whenever they detected that a horizontal line appeared in the tactile stimulator [17].

**Electrophysiology.** A custom-designed electrode Neuroscan cap, and an ATI electroencephalography (EEG) system were used to record high-density (64 channels) EEG signals during tactile stimulation [17, 37]. Furthermore, impedances were kept underneath 10kOhms. We included additional electrodes to monitor eye movement (superior and inferior orbits and the left and right lateral canthi) and references (bilateral mastoids) [38]. Then, as in previous studies [16], the raw signals were digitized using a sampling frequency of 1000 Hz. After the acquisition was conducted, data were analyzed through a band-pass filter of 0.05–30 Hz and a 50 Hz notch filter. The detection of muscle contractions and eye movement artifacts was immediately achieved by visually observing the EEG waves. Hence, muscle and eye movement artifacts were visually identified offline on a trial-by-trial basis and were removed prior to ERP, average data processing, and ERP analysis. Also, noisy channels were replaced with moderate linear interpolations of adjacent clean channels [38].

The averages for each participant in the remaining artifact-free trials were calculated. A temporal recording window (also known as epochs) lasting 1000 ms was opened, such that the recording started 100 ms before stimulus presentation and continued 900 ms after stimulus presentation, including 300 ms stimulus. In addition, the average voltage for 100 ms prior to stimulus onset was defined as the baseline. The analysis of the EEG was conducted on frequent

stimuli (non-target trials) to avoid contamination from motor-related neural activity associated with making a motor response [16, 17].

**Source localization reconstruction.** Regarding the cerebral source localization, the sources of the P100 wave were estimated from the 64-electrode recordings in all participants. The EEG inverse problem was applied using the Bayesian Model Averaging (BMA) approximation [39] to localize the sources of the ERP. Meanwhile, Low-Resolution Electromagnetic Tomography (LORETA) [40] was used to decipher the individual models. Moreover, each model was defined by constraining the solution to a specific anatomical structure using statistical mapping (SPM8) software [41, 42]. Some structures (18 areas from the cerebellum and eight areas that comprised less than 10 voxels) were excluded from consideration. The P100 wave generated was analyzed 80–120 ms after the trigger and opening time window of -20 to +20 ms, starting from the highest positive amplitude peak measured using the PZ electrode, to perform the BMA analysis at non-target stimuli (vertical lines).

### Statistics

The Shapiro–Wilk test was used to verify the data normality. Data were expressed as frequency, mean ±standard deviation, and median [25th and 75th percentile]. After that, the demographics between groups were compared using one-factor analysis of variance (ANOVA) or chi-squared test ($x^2$), where appropriate. In addition, a Student's t-test or U of Mann-Whitney's test was used to compare EB and LB groups in terms of clinical visual-related data, as appropriate. P100 latencies between the three groups were compared using a Kruskal–Wallis' test. The statistical significance was set at $p<0.05$, and data analysis was performed using IBM SPSS Statistics software (v.23.0; IBM Corp, Armonk, NY, USA).

The statistically significant sources within each group and between groups were determined using the SPM8 software (The MathWorks Inc., Natick, MA, USA) Thus, an analysis of independent Hotelling's T2 test against zero was first developed within each group to see the areas of greatest group activity [39, 43]. Then, the differences between groups were analyzed to determine the areas that differed the most during the passive tactile recognition task, which computation was based on a voxel by voxel independent Hotelling's T2 test. The SPM was used to make population inferences over the calculated sources of the P100 wave. The software mentioned above displayed the statistically significant source, and the significance level was set at $p<0.05$. For the between groups comparisons, the statistical analyses were conducted in accordance with the comparisons between the three groups, with the probability of type I error set at 0.017. This was calculated by dividing the established error level (0.05) by the number of comparisons (3).

The sample size was determined using the free software G*Power (v. 3.19.4; Heinrich-Heine-Universität Düsseldorf, Germany), considering the information obtained from previous research for the P100 latency variable [17]. A sample of twelve participants per group was required for setting an alpha risk of 0.05 and a beta risk of 0.1 in a bilateral contrast. Finally, 27 participants with a VI (14 EB and 13 LB) and 15 controls were included in the study.

### Results

During the participant recruitment period, fifty-four people with VI responded to the call to participate in this study. 27 of them met the inclusion criteria (14 EB individuals and 13 LB subjects). Regarding the control group, 19 people were recruited, but only 15 participated in the study.

The latencies, demographical and clinical data were obtained from 42 participants, while the localization source was acquired from 41 subjects. As a result, the source localization from

one individual in the LB group was not properly recorded due to technical problems, and it was not processed in the final analysis.

## Demographic and clinical data

The EB group consisted of 14 individuals (nine women and five men) with a mean age of 41.14 ± 11.09 years. Thirteen subjects (eight women and five men) belonged to the LB group (mean age of 43.15 ± 11.98 years). In addition, the CG consisted of 15 individuals (seven women and eight men) with a mean age of 40.47 ± 11.04 years. The group members predominantly had university-level education (50% for EB, 76.9% for LB, and 80% for CG). There were no differences between groups regarding age (p = 0.700), sex (p = 0.633), and educational level (p = 0.216).

The leading causes of blindness were glaucoma, premature retinopathy, retinitis pigmentosa, retrolental fibroplasia, and optic nerve atrophy. As displayed in Table 1, there were no differences between EB and LB groups regarding the following variables: type of VI, carrying out a rehabilitation program, time spent on that program, and devices used to tackle their daily mobility. However, the EB group has lived longer with blindness (40.07 ± 10.557 years) than the LB group (15.08 ± 10.340 years, p<0.001).

## P100 latency

After analyzing the first positive wave (P100) of the ERPs, we observed that the P100 latency was lower in the EB individuals (90.5 ms [74.0;104.5]), than in the LB (100.0 ms [95.0;111.5]), and CG (95.0 ms [70.0;106.0]) groups. Nonetheless, the three groups had no statistically significant differences in P100 latencies ($H(2) = 3.192$, p = 0.203) (Fig 2).

## P100 source localization

**Source localization in each group.** During the tactile horizontal line recognition task, several cortical regions showed a higher statistically significant activation. The area with

**Table 1. Clinical characteristics of the visual impaired individuals.**

| Variable | EB (n = 14) | LB (n = 13) | P |
|---|---|---|---|
| **Type of VI n (%)** | | | |
| Blindness | 11 (78.5) | 10 (76.9) | 1.000[a] |
| Severe visual deficit | 3 (21.5) | 3 (23.1) | |
| **VI time (years)** | 40.07 ± 10.56 | 15.08 ± 10.34 | **< 0.001[b]** |
| **Rehab. Prog. n (%)** | | | |
| Yes | 11 (80.0) | 11 (78.6) | 1.000[a] |
| No | 3 (20.0) | 2 (21.4) | |
| Rehab. (months) | 5.50 [1.00;8.25] | 3.00 [1.00;5.00] | 0.280[c] |
| **Mobility n (%)** | | | |
| White cane/guide dog | 13 (93.3) | 11 (78.6) | 0.596[a] |
| Seeing guide | 1 (6.7) | 2 (21.4) | |

Data are presented as mean ± standard deviation; median and interquartile range [percentile 25; percentile 75] (Mdn [RIC]) or as frequency and percentage (n (%)). Statistically significant differences are highlighted in bold. VI: Visual impairment; EB: early-onset blindness; LB: late-onset blindness; Rehab. Prog: rehabilitation program. It was considered as statistically significant when p<0.05.

[a]Association between variables calculated with Fisher's exact test. (Bilateral signification).

[b]Contrast carried out with the T-Student's test.

[c]Contrast carried out with the U of Mann-Whitney's test.

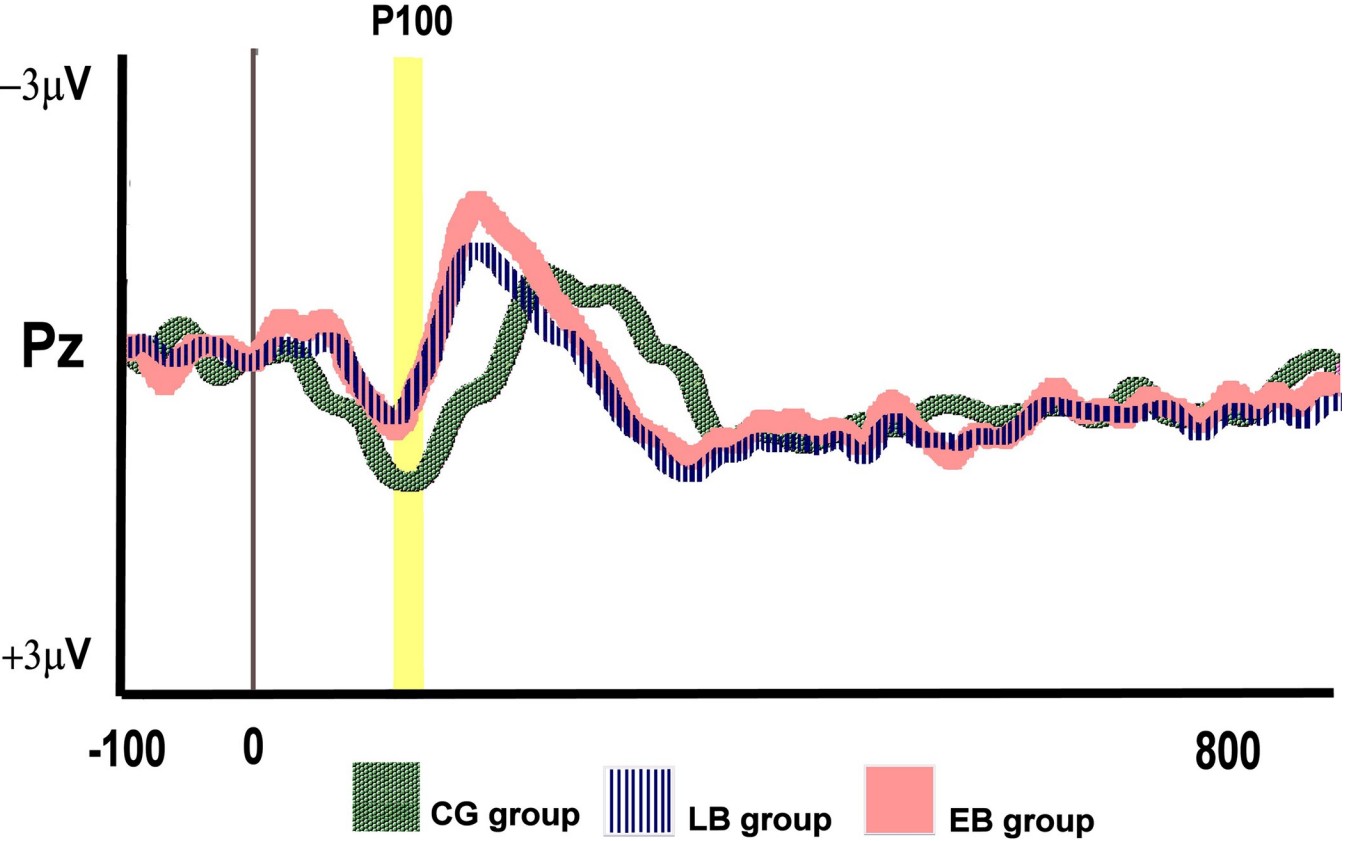

**Fig 2. Event-related potentials waves are observed at the Pz electrode in all groups.** The P100 wave was determined by locating the maximum amplitude in the respective time window using the Pz electrode. EB: early-onset blindness; LB: late-onset blindness; CG: control group.

maximal activation in CG individuals was located at the right frontal superior lobe, while for EB subjects, the maximal activity was found in the bilateral superior, medial frontal, and left occipital regions. In the LB group, the following areas showed a higher statistically significant activity during the task above: bilateral superior and medial frontal areas, orbitofrontal zone, and bilateral anterior cingulate cortex (Fig 3 and Table 2).

**Differences in source localization between groups.** At the peak amplitude of P100 during the tactile horizontal line recognition task, the EB group exhibited a higher statistically significant activity compared to the CG in right superior parietal zones. On the other hand, the LB group showed higher statistically significant activation in left orbitofrontal areas compared to the CG. Besides, EB subjects demonstrated statistically significant maximal activation in the left orbitofrontal regions when compared to LB individuals (Fig 4 and Table 3).

## Discussion

The results of this study indicated that, contrary to the initial hypothesis, P100 latencies did not differ significantly between EG, LB, and CG subjects. Previous research found differences in P100 latency between VI individuals and sighted subjects [17, 19]. According to Collignon and De Volder [11] not only is the improved performance of VI subjects due to intermodal neuroplastic compensations, but it is also the result of the enhanced efficacy of top-down mechanisms. Attention is a prerequisite for perception [44], and the P100 wave reveals how data is transferred from the stimulus to the consciousness [45]. However, it is well-known that

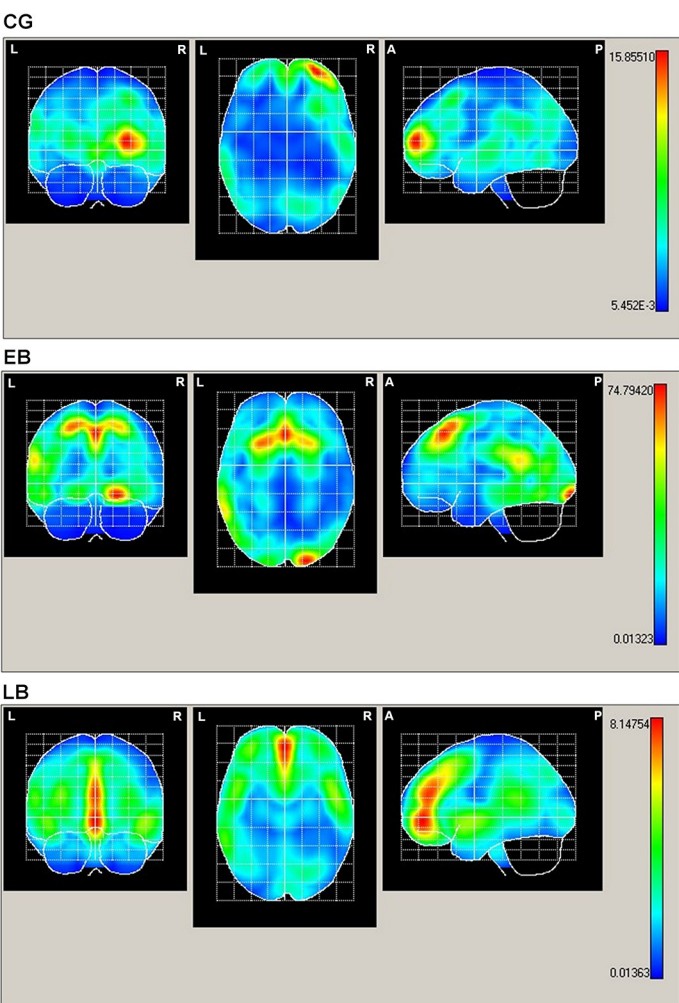

**Fig 3. Horizontal coronal and mid-sagittal view of the mean projection of the brain areas activated for the P100 wave to tactile stimuli of the horizontal line.** Summary of brain areas of maximum activation. EB: early-onset blindness; LB: late-onset blindness; CG: control group.

the P100 wave is not attributed to voluntary control of attention [24] (e. g., selective or divided attention), but is related to early attention processing [46]. This means that the P100 wave arises from low-level cortical regions, allowing the passage of input activity to higher-order stages. In fact, regardless of the paradigm used, it is involved in initial bottom-up pattern processing [24]. Therefore, this clarifies why differences in P100 latencies between groups were not detected, as it seems that all the study subjects had high early attention, since they performed the task for the first time. In contrast, participants of other studies where latencies were shorter in the VI group had previous experience with the tactile task [17, 19].

Regarding the source localization in each group, we found that the three groups activated regions of their frontal lobes. This may be due to the use of the attention and working-memory processes required for the tactile recognition of the stimulus presented. This is consistent with our P100 latency findings. Moreover, the frontal lobe controls decision-making and executive function through attention and memory processes [47–49].

All participants showed a higher statistically significant activation of the right superior frontal regions, which control impulses. In addition, the superior frontal gyrus participates in

**Table 2. Main neuroanatomical structures according to projection areas of maximal intensity in P100 wave.**

| GROUP | aal | X | Y | Z | Hotelling's $T^2$ |
|---|---|---|---|---|---|
| CG | Frontal Superior R | 30 | 62 | 8 | 15.5 |
| EB | Lingual R | 22 | -94 | 12 | 74.79 |
| | Frontal Superior Medial R | 6 | 27 | 56 | 60.23 |
| | Frontal Superior Medial L | -3 | 28 | 56 | 61.64 |
| | Frontal Superior R | 21 | 22 | 54 | 54.15 |
| | Frontal Superior L | -20 | 22 | 55 | 62.86 |
| LB | Frontal Middle Orbital L | -2 | 50 | -4 | 8.14 |
| | Frontal Middle Orbital R | 3 | 53 | -4 | 7.81 |
| | Frontal Superior Medial L | -2 | 44 | 22 | 7.64 |
| | Frontal Superior Medial R | 3 | 50 | 7 | 7.11 |
| | Anterior Cingulum L | -2 | 43 | 20 | 7.53 |
| | Anterior Cingulum R | 2 | 50 | 17 | 7.14 |

aal: Anatomical label corresponding to probabilistic brain atlas. X, Y, Z: coordinates from aal in three spatial axes. $T^2$: Hotelling's $T^2$ test. CG: control group; EB: early-onset blindness; LB: late-onset blindness. R: right; L: left. It was considered as statistically significant when $p < 0.05$.

inhibitory impulse control and motor urgency [50] and inhibitory top-down process [51]. Notably, our participants had to press the button only after recognizing the horizontal lines, so they had to control themselves. Furthermore, it has been observed that alterations in the right cortical frontal zones could lead to modifications in the connectivity of this area, which are linked to changes in impulse control [52].

Only the VI groups showed a higher statistically significant activation of the left superior frontal areas. The function of the left superior gyrus is to process spatial information, which is also key for a working memory [53, 54]. The tactile recognition task in our study was based on the detection of tactile stimuli with a given spatial orientation, which is consistent with activation of the left superior frontal area. In addition, a recent study demonstrated that children with a VI have more enhanced working-memory than sighted children [55]. These findings were also reported in adults with a VI compared with sighted adults [56].

Attention should be given to the higher statistically significant activation of the mid-frontal areas in VI participants. Notably, the mid-frontal gyrus is a mediational structure between the dorsal and ventral attentional networks [57], and the right mid-frontal gyrus functions as a probabilistic area inside the dorsolateral prefrontal cortex. The latter is responsible for sustained attention and self-control [58, 59], while the left mid-frontal gyrus participates in selective attention [57]. Previous studies have found activation in these areas in people with VI when recognizing vibrotactile stimuli [17, 60, 61]. This may be because VI individuals require

**Table 3. Main neuroanatomical structures according to projection areas of maximal intensity in P100 wave between groups.**

| Group | aal | X | Y | Z | Hotelling's $T^2$ | CG<EB |
|---|---|---|---|---|---|---|
| CG vs. EB | Parietal Superior R | 15 | -78 | 52 | 11.17 | |
| Group | aal | X | Y | Z | Hotelling's $T^2$ | CG<LB |
| CG vs. LB | Frontal Inferior Orbital L | -39 | 41 | -16 | 11.85 | |
| Group | aal | X | Y | Z | Hotelling's $T^2$ | LB<EB |
| LB vs. EB | Frontal Inferior Orbital L | -42 | 37 | -16 | 24.7 | |

Anatomical label corresponding to probabilistic brain atlas. X, Y, Z: coordinates from aal in three spatial axes. T2: Hotelling's T2 test. CG: control group; EB: early-onset blindness; LB: late-onset blindness. R: right; L: left. It was considered as statistically significant when $p < 0.017$.

*CG vs. EB*

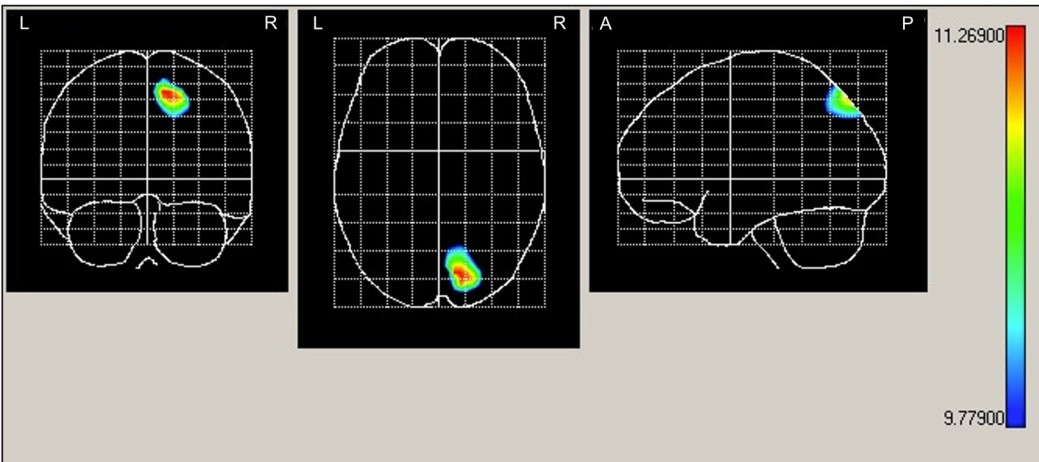

*CG vs. LB*

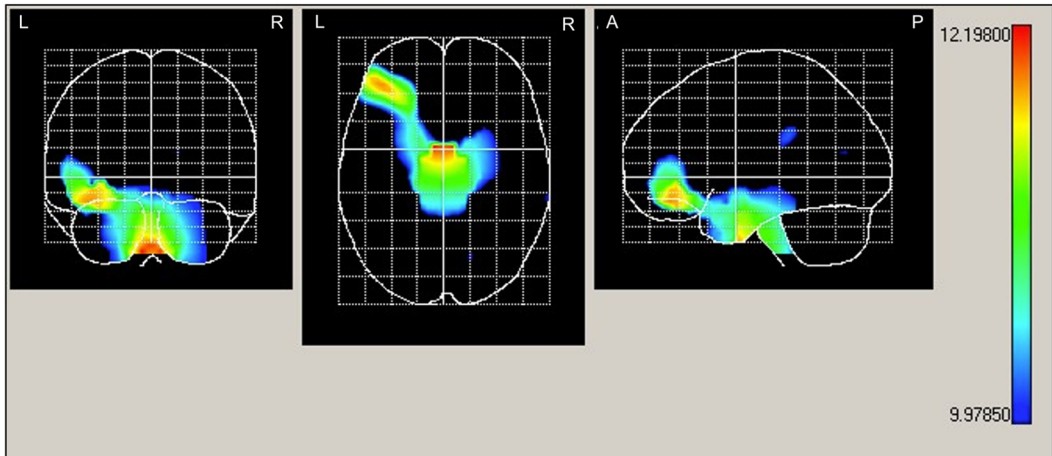

*LB vs. EB*

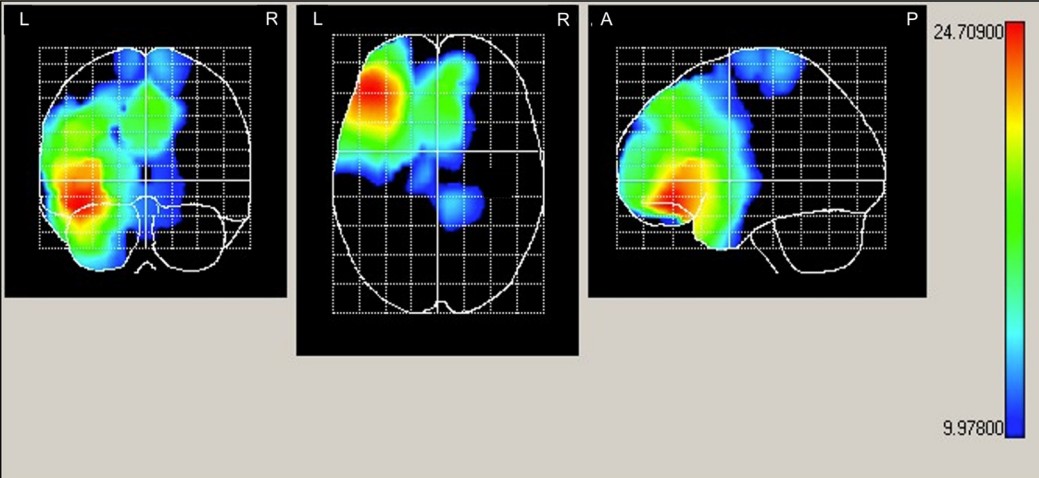

**Fig 4. Horizontal coronal and mid-sagittal view of the cortical projections of intergroup comparisons.** Significant differences maps between groups are displayed in red. EB: early-onset blindness; LB: late-onset blindness; CG: control group. R: right; L: left. A: Anterior. P: posterior.

higher neuronal and neurochemical resources when conducting a specific task due to the lack of vision [33, 34, 62], which implies a higher attentional demand [17].

A higher statistically significant activity appeared in the anterior cingulate cortex and orbitofrontal areas of LB subjects. The higher activity of the orbitofrontal area was also found in LB individuals when compared to CG. During decision-making, the anterior cingulate cortex plays an essential role in error detection, outcome tracking, and evaluating positive or negative results [63]. Additionally, the anterior cingulate and the orbital cortex are synchronized during an effort-based decision-making task [64]. Burton et al. [65] detected activation in areas close to the anterior cingulate cortex in LB people, suggesting the participation of emotional processes during the task. Moreover, the orbitofrontal cortex is involved in reward value and emotional processing. This structure then sends data to the anterior cingulate cortex to learn from the reward or non-reward results [66, 67].

On the other hand, the higher statistically significant activation found in the occipital areas of EB participants in our study, absent in the LB group, suggests that this findings could be due to cross-modal plasticity and its variations, depending on the age of onset blindness. Cross-modal plasticity mainly consists of the brain's capability to perceive and process stimuli from a sensorial modality different from the original input [13, 68]. It is an adaptive form of neuroplasticity that appears when there is restriction or loss of a sensorial channel [68]. This has been well-documented in people with VI [12, 13, 68].

It is well-documented in the scientific literature that the occipital cortex remains activated even after blindness acquisition [17, 34, 69]. Therefore, the activation of occipital structures in EB subjects in our research may be because they mainly utilize the forenamed regions to manage non-visual information. In fact, some studies have claimed that in early-onset visual deprivation, the specialization of hetero-modal areas is preserved, and the ventral and dorsal visual streams are adapted to managing auditory and tactile data [34, 70]. Notably, the LOC is a structure that processes non-visual stimuli and, as Voss [70] and Ortiz Alonso [17] stated, it is active in EB individuals, suggesting no need for prior visual experience to recognize tactile objects and shapes in the occipital cortex.

In addition, the visual cortex participates in the early recognition of tactile objects (usually between 100–150 ms post-stimulus) [23, 71], which is consistent with the present study's results from the recognition of horizontal lines task. However, a special mention should be made of the fact that no statistical significant higher activity was found in the occipital region in the LB group. According to Lazzouni and Leporé [34], this could be due to the fact that the occipital areas of people with LB remains preserved for visual information management.

Finally, our results indicate that, compared to the sighted, the EB group exhibited greater activity in the right superior parietal region. Superior parietal areas are considered as structures that manage a wide range of somatosensory and cognitive processes (e.g., somatosensory integration, mental rotation, motor learning, spatial perception, etc.) [72], and their volume and connectivity are increased in people with EB [73, 74]. Our results may be explained because, after early visual loss, the parietal region would consequently acquire the capacity to integrate a greater amount of multimodal information compared to sighted individuals [75]. This augmented capacity could also result from a greater tactile experience [34], as people with VI typically interact with the external world through their sense of touch [76].

It is important to note the greater activity observed in the left orbitofrontal area among participants with EB compared to LB individuals. The increased activation of this region in the EB group could be linked to the early lack of vision, influencing emotional processes, decision-making, social behavior and cognitive flexibility in a more profound manner. The orbitofrontal cortex plays a crucial role in the ability to adapt flexibly to a dynamically changing environment [77]. In the present study, vertical and horizontal lines where presented in a random

sequence, and participants were required to press the button only when the horizontal line appeared, thereby complying with the examiner's instructions. The maximum activation observed in the orbitofrontal cortex of subjects with EB aligns with the forenamed line detection sequence, as this cortical region is associated with the processing of emotional signals that guide human behavior in tasks where social judgment, reward and punishment are present [66, 67]. Additionally, the orbitofrontal cortex is implicated in regulating social behavior, including solving interpersonal problems and fostering social closeness [78]. Vision loss may prompt behavioral adaptations in social interactions (e. g., politeness, humor, avoidance, apologies, etc.) [79], enhanced decision-making [80] and improved detection of ecologically significant emotions [81].

## Limitations and future research

There are some limitations to this research that should be noted. Specifically, as mentioned in previous sections, we could not analyze the data from one participant due to technical problems. Although there are no apparent reasons why this could have affected our findings, it should be considered. In addition, it is necessary to recognize the difficulties in generalizing this study's results to the entire VI population. The inclusion criteria determined that participants should have total blindness or a severe visual deficit acquired before five or after 14 years old, which led them to use technical aids for mobility. Therefore, the study sample represents a distinct group of individuals evaluated at a specific moment.

Furthermore, it should be noted that, as previously stated, the P100 wave is modulated by unconscious and involuntary attention in paradigms where the subject must direct attention to a spatial stimulus [24]. However, the analysis of executive responses associated with deviant stimuli is more closely related to cognitive and decision-making processes [82], which were not the subject of this study and therefore this aspect was not examined. Future studies should focus on the analysis of the deviant stimulus to gain insight into the cognitive and decision-making processes of people with VI.

## Conclusions

This study provides insights into the neuroplastic mechanisms underlying visual deprivation. The EEG brain activity displayed the participation of the right superior frontal structures during the task in all the study's participants. Nonetheless, only the EB and LB subjects showed maximal activation of their left superior frontal areas, essential for working-memory and spatial processing. Additionally, the EB and LB participants activated mid-frontal areas that are relevant for attentional processes. On the other hand, LB subjects activated the anterior cingulate cortex and orbitofrontal areas, suggesting emotional processing and a task reward evaluation. The higher activation of the orbitofrontal cortex in LB was also found when compared to sighted individuals. Moreover, EB individuals showed a higher statistically significant activation of the occipital areas due to these regions' role in managing non-visual data. Participants with EB showed maximal activity in the superior parietal region compared to sighted subjects. In addition, when compared to the LB group, individuals with EB exhibited greater activity in the left orbitofrontal region. These findings suggest multimodal information processing, as well as emotional processing associated with social behavior regulation and reward elicitation.

Furthermore, variations in P100 latencies between participants with VI and sighted individuals were not detected in this study. This suggests that people with a VI may not have faster processing of early attention. However, whether their superior abilities are attributed to faster cognitive processing remains unknown. Therefore, further studies are required to clarify the participation of attentional processes in people with a VI. Ultimately, this knowledge may

assist clinicians and physiotherapists in enhancing rehabilitation programs to foster compensatory strategies in this population. These findings may also affect the development of appropriate neuroprostheses to aid or restore vision.

## Supporting information

**S1 Checklist. STROBE checklist.**
(DOCX)

## Acknowledgments

The authors would like to thank the collaboration of Dr. Rosa Henche, Mónica Inmaculada Fuster-Ribera, Dr. Ramón Ahulló Hermano and Dr. Álvaro García López for their kind assistance and technical help during the ERP recording. We would also thank the collaboration of Dr. María del Carmen Bravo for their help during the statistical analysis. The authors thank the participants for the contribution to the study.

## Author Contributions

**Conceptualization:** Mónica-Alba Ahulló-Fuster, M. Luz Sánchez-Sánchez, Tomás Ortiz.

**Data curation:** Tomás Ortiz.

**Formal analysis:** Mónica-Alba Ahulló-Fuster, M. Luz Sánchez-Sánchez, Tomás Ortiz.

**Funding acquisition:** Mónica-Alba Ahulló-Fuster.

**Investigation:** Mónica-Alba Ahulló-Fuster, Enrique Varela-Donoso, Tomás Ortiz.

**Methodology:** Mónica-Alba Ahulló-Fuster, Tomás Ortiz.

**Resources:** Mónica-Alba Ahulló-Fuster, Tomás Ortiz.

**Supervision:** Enrique Varela-Donoso, Tomás Ortiz.

**Writing – original draft:** Mónica-Alba Ahulló-Fuster.

**Writing – review & editing:** Mónica-Alba Ahulló-Fuster, M. Luz Sánchez-Sánchez, Enrique Varela-Donoso, Tomás Ortiz.

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
