## [Decision Letter · Decision Letter 0]

31 Oct 2023

PONE-D-23-21188Early attentional processing and cortical remapping strategies of tactile stimuli in adults with an early and late-onset visual impairment: A cross-sectional studyPLOS ONE

Dear Dr. Sánchez-Sánchez,

Thank you for submitting your manuscript to PLOS ONE. After careful consideration, we feel that it has merit but does not fully meet PLOS ONE’s publication criteria as it currently stands. Therefore, we invite you to submit a revised version of the manuscript that addresses the points raised during the review process.

Two experts carefully revised the manuscript. My main concern reflects the one of Reviewer#1 about the methodological aspects. This study, although it is very fascinating, does not reach methodological rigour, particularly regarding the statistical analyses. The authors should follow the suggestions provided to improve their manuscript.

We look forward to receiving your revised manuscript.

Kind regards,

Valentina Bruno

Academic Editor

PLOS ONE

5. We note that Figure 1 in your submission contain copyrighted images. All PLOS content is published under the Creative Commons Attribution License (CC BY 4.0), which means that the manuscript, images, and Supporting Information files will be freely available online, and any third party is permitted to access, download, copy, distribute, and use these materials in any way, even commercially, with proper attribution. For more information, see our copyright guidelines: http://journals.plos.org/plosone/s/licenses-and-copyright.

Reviewers' comments:

Reviewer's Responses to Questions

**Comments to the Author**

1. Is the manuscript technically sound, and do the data support the conclusions?

Reviewer #1: No

Reviewer #2: Yes

2. Has the statistical analysis been performed appropriately and rigorously? 

Reviewer #1: No

Reviewer #2: Yes

3. Have the authors made all data underlying the findings in their manuscript fully available?

Reviewer #1: Yes

Reviewer #2: Yes

4. Is the manuscript presented in an intelligible fashion and written in standard English?

Reviewer #1: Yes

Reviewer #2: No

5. Review Comments to the Author

Reviewer #1: This study focuses on the very interesting topic of cross-modal plasticity, utilizing individuals with visual impairment, comparing early-onset and late-onset blindness to sighted individuals. Using a spatial tactile task and event-related potentials, the authors found no significant differences in early attentional processing (P100 latency) among the groups. The study reports that early-onset blindness activated the occipital lobe, while late-onset blindness showed activation in the orbitofrontal area, suggesting emotional processing differences in response to tactile stimulation. It is a very well-planned study up until the point of data handling and analysis. However, the main argument of the authors focuses on source activity differences, (as ERP differences were not found) but again this is not supported by an appropriate statistical analysis. Below I report my main concerns for the study:

1. My biggest concern regards the absence or insufficient reporting of source activity statistical analysis. Specifically, the authors report that they used SPM to perform the analysis, but from my understanding the analysis was performed within each group via a one sample test. This approach cannot reveal group differences, though. If the authors want to ground an argument in different loci of activation, they should perform a between group comparison with 3 groups, via an F test if they have two-directional hypotheses or a T test if they have one-directional hypotheses. That would actually show if the different groups utilized significantly different cortical patterns of activation to process the stimuli. The regions involved should be discussed and interpreted in such an analysis.

In relation to the above, my suggestion would actually be to model the responses to the deviant stimuli as well and include them in the analysis via a 2x3 mixed model design with between subjects factor group and within subjects factor condition. There are several reasons why such an analytical approach would better answer the crucial research question set by the authors:

1.1. The early attentional effects on P100 are better evidenced when comparing conditions, not when the P100 is compared against zero, as the inter-individual differences in the amplitude of the ERP (affected by variables such as the scalp thickness etc) do not allow such an approach to show valid estimation of attentional effects modulation.

1.2. I understand that the authors would like to avoid the deviant condition due to motor artifacts, but if they focus on P100 and not later components, not even preparatory action planning processes would be visible within this time-range (the typical time range is 80-120ms, participants cannot be that fast in evaluating the stimulus and already start preparing for action). Even later components up until 250 ms would be safely analyzed in that way.

Overall a significant interaction of condition x group in such an analysis would constitute a valid index of differences in cortical responses related with cross-modal plasticity and early attentional effects.

2. Report of ERP and statistical analysis is not complete. The authors report that the latency is not significantly different amongst groups and provide the corresponding statistical results. Nonetheless, they report that “these waves were similar for the cerebral responses in the three groups”, and similarity is not evaluated somehow, nor the very important measure of ERPs’ amplitude. If a statistically significant effect exists in the amplitude of the P100, it may ground the argumentation provided in the discussion. If no statistically significant effect exist, then basically the authors have an interesting outcome which if not caused by other confounding variables would call for a discussion of why cross-modal plasticity was not present in the study.

3. The main argumentation presented in the discussion expands on possible cross-modal processing differences between the groups. Nonetheless, such differences were not reported statistically in the paper. Hence, the authors would either perform other (and probably more appropriate) analyses to show the effect, or actually discuss more an interpretation of why differences in haptic processing were not observed amongst the groups.

4. As a minor point, the authors should add a description of P100, how it is modulated by early attentional processes and what are its generators in the introduction so that the reader understands why they decided to use this as an index of the corresponding effects, and whether the interpretation for their results regarding source estimation are reasonably connected to the corresponding literature (the latter should be added in the discussion section).

Reviewer #2: Page 5, Methods: (line 109, 114): For early-onset blindness, the authors have considered loss of vision before five years of age. The cutoff age is from 0 to 5 years, with many studies limiting to less than 2 years.

What was the rationale?

Page 7, Line 166: Why was the sampling frequency kept at only 1000 Hz?

Elaborate on the rationale behind epoch between 100ms pre-stimulus to 900ms post.

Page 14, Line 294: “Therefore, these findings suggest the possible existence of a critical period for improving the attentional mechanisms that follow vision loss.”

The authors should consider making an informed statement from the literature on this. Since the age of the subjects at the time of study is above 40 years, the inference is incorrect.

Page 16, line 344: Regarding occipital areas, the authors comment “variations occur in cross modal plasticity and depends on the age of onset blindness.”

It seems a general statement based on the literature, rather than their conclusion. The authors have not associated the details regarding any association between the two. They may add this information.

6. PLOS authors have the option to publish the peer review history of their article (what does this mean?). If published, this will include your full peer review and any attached files.

Reviewer #1: **Yes: **Evangelos Paraskevopoulos

Reviewer #2: **Yes: **S Senthil Kumaran

---

## [Author Response · Author response to Decision Letter 0]

9 Jan 2024

We thank the editor and the reviewers (1 and 2) for their critical review and valuable comments. We have taken into account all of their recommendations and suggestions. Itemized responses are listed below. All the modifications have been clearly highlighted using the "Track Changes" throughout the manuscript to make its revision easier. 

Academic editor:

We have carefully reviewed the style requirements.

2. Did you know that depositing data in a repository is associated with up to a 25% citation advantage (https://doi.org/10.1371/journal.pone.0230416)? If you’ve not already done so, consider depositing your raw data in a repository to ensure your work is read, appreciated and cited by the largest possible audience.

We have deposited data in a repository (https://doi.org/10.7910/DVN/K7IGAB).

3. In your Data Availability statement, you have not specified where the minimal data set underlying the results described in your manuscript can be found. PLOS defines a study's minimal data set as the underlying data used to reach the conclusions drawn in the manuscript and any additional data required to replicate the reported study findings in their entirety.

We have deposited data in a repository. Relevant URL (https://doi.org/10.7910/DVN/K7IGAB) has been included in our revised cover letter.

This issue has been fixed.

5. We note that Figure 1 in your submission contain copyrighted images. All PLOS content is published under the Creative Commons Attribution License (CC BY 4.0), which means that the manuscript, images, and Supporting Information files will be freely available online, and any third party is permitted to access, download, copy, distribute, and use these materials in any way, even commercially, with proper attribution.

Thank you. Figure 1 was created specifically for this study by a collaborator who is listed in the Acknowledgments section. You can contact him if you need confirmation: Ramon Ahulló. E-mail: ramon.ahullo@uv.es 

Reviewers' comments:

Reviewer 1: 

1. My biggest concern regards the absence or insufficient reporting of source activity statistical analysis. Specifically, the authors report that they used SPM to perform the analysis, but from my understanding the analysis was performed within each group via a one sample test. This approach cannot reveal group differences, though. If the authors want to ground an argument in different loci of activation, they should perform a between group comparison with 3 groups, via an F test if they have two-directional hypotheses or a T test if they have one-directional hypotheses. That would actually show if the different groups utilized significantly different cortical patterns of activation to process the stimuli. The regions involved should be discussed and interpreted in such an analysis. In relation to the above, my suggestion would actually be to model the responses to the deviant stimuli as well and include them in the analysis via a 2x3 mixed model design with between subjects factor group and within subjects factor condition. There are several reasons why such an analytical approach would better answer the crucial research question set by the authors:

1.1. The early attentional effects on P100 are better evidenced when comparing conditions, not when the P100 is compared against zero, as the inter-individual differences in the amplitude of the ERP (affected by variables such as the scalp thickness etc.) do not allow such an approach to show valid estimation of attentional effects modulation.

Answer: We appreciate the reviewer's comment and agree with the reviewer's suggestion. In this sense, the differences between the three groups in the increased activity of brain areas were analyzed. We believe that the results are now consistent with the aims of the study. The differences between groups were analyzed using Hotelling’s T2 because, as stated in the literature, the analysis of electroencephalogram signals challenges traditional statistics in several ways (Carbonell et al., 2004; Trujillo-Barreto et al., 2004); thus, Hotelling’s T2 has been proposed for electrophysiological (EEG) neuroimaging analyses (Carbonell et al., 2004) and has previously been used to analyze differences between groups (Requena & López, 2014; Li & Yang, 2022). In addition, some authors have pointed out that Hotelling’s T2 test has a significantly lower type 1 error rate compared to the ANOVA F-test (Lopez, 2019). 

Therefore, thanks to the reviewer's comments, some information has been added to the Results and Statistical Analysis sections. 

Please, see lines 234-241: The statistically significant sources within each group and between groups were determined using the SPM8 software (The MathWorks Inc., Natick, MA, USA). Thus, an analysis of independent Hotelling’s T2 test against zero was first developed within each group to see the areas of greatest group activity [39,43]. Then, the differences between groups were analyzed to determine the areas that differed the most during the passive tactile recognition task, which computation was based on a voxel by voxel independent Hotelling’s T2 test. 

Please, see lines 316-321: At the peak amplitude of P100 during the tactile horizontal line recognition task, the EB group exhibited a higher statistically significant activity compared to the CG in right superior parietal zones. On the other hand, the LB group showed higher statistically significant activation in left orbitofrontal areas compared to the CG. Besides, EB subjects demonstrated statistically significant maximal activation in the left orbitofrontal regions when compared to LB individuals (Figure 4 and Table 3). 

References: 

Carbonell, F., Galán, L., Valdés, P., Worsley, K., Biscay, R. J., Díaz-Comas, L., Bobes, M. A., & Parra, M. (2004). Random field-union intersection tests for EEG/MEG imaging. NeuroImage, 22(1), 268–276. https://doi.org/10.1016/j.neuroimage.2004.01.020

Li X, Yang L, Yan X. An exploratory study of drivers' EEG response during emergent collision avoidance. J Safety Res. 2022 Sep;82:241-250. doi: 10.1016/j.jsr.2022.05.015.

Lopez, Jay, "Effect of Heterogeneity of Variance on the performance of ANOVA F-test and its Alternatives: Simulation Based Study" (2019). University Honors Theses. Paper 788. https://doi.org/10.15760/honors.806

Requena C, López V. Measurable benefits on brain activity from the practice of educational leisure. Front Aging Neurosci. 2014 Mar 11;6:40. doi: 10.3389/fnagi.2014.00040. PMID: 24653699; PMCID: PMC3949114

Taylor J. G. (2007). On the neurodynamics of the creation of consciousness. Cognitive neurodynamics, 1(2), 97–118. https://doi.org/10.1007/s11571-006-9011-8

Trujillo-Barreto, N. J., Aubert-Vázquez, E., & Valdés-Sosa, P. A. (2004). Bayesian model averaging in EEG/MEG imaging. NeuroImage, 21(4), 1300–1319. https://doi.org/10.1016/j.neuroimage.2003.11.008

1.2. I understand that the authors would like to avoid the deviant condition due to motor artifacts, but if they focus on P100 and not later components, not even preparatory action planning processes would be visible within this time-range (the typical time range is 80-120ms, participants cannot be that fast in evaluating the stimulus and already start preparing for action). Even later components up until 250 ms would be safely analyzed in that way. Overall a significant interaction of condition x group in such an analysis would constitute a valid index of differences in cortical responses related with cross-modal plasticity and early attentional effects.

Answer: We appreciate the reviewer for his/her comment. In this regard, we agree with the reviewer’s perspective that if we had aimed to assess more complex cognitive processes such as working-memory, an analysis of late components (e.g., those starting at 250 milliseconds) would have been more suitable. However, we intended to observe whether there were changes in early attentional levels, for which the P100 component is an optimal waveform. In fact, as mentioned in the manuscript (lines 346-350), the P100 is considered to provide evidence of the first stages of neuronal processing in the neocortical circuit and it allows the passage of information from lower cortical stages to higher order stages. Therefore, it does not require “preparatory” or voluntary attention from the individual. (Taylor, 2007). Furthermore, P100 is a positive waveform peaking between 80-120 ms (Wynn et al., 2008). 

Therefore, thanks to the reviewer's comments, some information explaining P100 has also been added. Please, see lines 114-128: Somatosensory event-related potentials (ERPs) can reveal valuable data about processing spatial tactile information in the cortex [23]. After a stimulus is presented, the electroencephalogram typically exhibits a pattern of oscillations, including the P100 waveform, nestled amidst a series of peaks and troughs. The emergence of P100 activity is believed to occur in primary sensorial cortical areas, offering insight into the initial phases of the neuronal processing within the neocortical circuit [24–27]. There is growing consensus concerning the contribution of P100 waveform in the context of global cognitive processing, notably in situations involving attention. In paradigms in which attention must be directed to a stimulus in space, the ERP P100 has been observed to be modulated by early attentional activity [24]. Spatial attention modulates neural activity as early as 100 ms after stimulus presentation [28,29]. Top-down modulation of cortical activity during the early phases of perceptual processing influences subsequent working-memory management [30]. Thus, neuroplastic changes in VI and early attentional processing can be explored using tools such as ERPs along with quantitative electroencephalography [17]. 

References: 

Taylor J. G. (2007). On the neurodynamics of the creation of consciousness. Cognitive neurodynamics, 1(2), 97–118. https://doi.org/10.1007/s11571-006-9011-8

Wynn, J. K., Lee, J., Horan, W. P., & Green, M. F. (2008). Using event related potentials to explore stages of facial affect recognition deficits in schizophrenia. Schizophrenia bulletin, 34(4), 679–687. https://doi.org/10.1093/schbul/sbn047

2. Report of ERP and statistical analysis is not complete. The authors report that the latency is not significantly different amongst groups and provide the corresponding statistical results. Nonetheless, they report that “these waves were similar for the cerebral responses in the three groups”, and similarity is not evaluated somehow, nor the very important measure of ERPs’ amplitude. If a statistically significant effect exists in the amplitude of the P100, it may ground the argumentation provided in the discussion. If no statistically significant effect exist, then basically the authors have an interesting outcome which if not caused by other confounding variables would call for a discussion of why cross-modal plasticity was not present in the study.

Answer: We would like to thank the reviewer for his/her comment. In this regard, we would like to point out that amplitude is a parameter that reflects the voltage changes per second in an EEG and it is usually measured at each electrode (Bradley & Keil, 2012). Nonetheless, the methodology of our study (lines 220-223) collects the average of the microvolts in all of the electrodes during a time window of 40 ms (-20 ms before and +20 ms after) in order to generate a three-dimensional image that provides us with information about the highest brain activity in each of the brain regions (see Source localization reconstruction section). Thus, Figure 2 and Table 3 show the statistically significant differences in the brain activity in each of the groups. The methodology used in the present study is based on the research objective of observing differences in brain activity in the different groups.

References: 

Bradley, M., & Keil, A. (2012). Event-Related Potentials (ERPs). In Encyclopedia of Human Behavior. , 79-85. https://doi.org/10.1016/B978-0-12-375000-6.00154-3.

3. The main argumentation presented in the discussion expands on possible cross-modal processing differences between the groups. Nonetheless, such differences were not reported statistically in the paper. Hence, the authors would either perform other (and probably more appropriate) analyses to show the effect, or actually discuss more an interpretation of why differences in haptic processing were not observed amongst the groups.

Answer: We thank the reviewer for his/her comment. In this regard, we would like to clarify that we are not evaluating cross-modal plasticity, but rather we are assessing which regions showed maximal activity in early blind, late blind and sighted controls when passively stimulated (not haptic). The fact that we are mentioning cross-modal plasticity is because this is a type of neuroplasticity that arises in people with visual impairment, so that when they are tactilely stimulated, they show increased activity in the occipital region (Lazzouni & Leporé, 2014; Ortiz Alonso et al., 2015). Since the early visually impaired group was the one that showed greater activity in the occipital region, we think that this could be the result of cross-modal plasticity processes. However, reading the wording of the manuscript, we understand that what is written is confusing. Apologies for the confusing wording. 

Thanks to the reviewer's comment, some sentences have been modified: 

Please, see lines 23-25: Neuroplastic changes appear in people with visual impairment (VI) and they show greater tactile abilities. Improvements in performance could be associated with the development of enhanced early attentional processes based on neuroplasticity.

Please, see lines 96-98: Therefore, this region is a clear example of multimodal spatial information processing [2]. These findings reinforce the hypothesis of substitutive neuroplasticity in people with VI [13].

Please, see lines 106-107: It is suggested that neuroplasticity could be behind the superior tactile skills found in people with VI. 

Please, see lines 132-134: We also set out to analyze the maximum brain activity in each group and to compare the brain activity in subjects with early and late-onset VI and a group of sighted individuals. 

Please, see lines 414-422: On the other hand, the higher statistically significant activation found in the occipital areas of EB participants in our study, absent in the LB group, suggests that this findings could be due to cross-modal plasticity and its variations, depending on the age of onset blindness. Cross-modal plasticity mainly consists of the brain’s capability to perceive and process stimuli from a sensorial modality different from the original input [13,68,69]. It is an adaptive form of neuroplasticity that appears when there is restriction or loss of a sensorial channel [69]. This has been well-documented in people with VI [12,13,69]. 

Please, see lines 423-434: It is well-documented in the scientific literature that the occipital cortex remains activated even after blindness acquisition [17,34,70]. Therefore, the activation of occipital structures in EB subjects in our research may be because they mainly utilize the forenamed regions to manage non-visual information. In fact, some studies have claimed that in early-onset visual deprivation, the specialization of hetero-modal areas is preserved, and the ventral and dorsal visual streams are adapted to managing auditory and tactile data [34,71]. Notably, the LOC is a structure that processes non-visual stimuli and, as Voss [71] and Ortiz Alonso [17] stated, it is active in EB individuals, suggesting no need for prior visual experience to recognize tactile objects and shapes in the occipital cortex. 

References: 

Lazzouni, L., & Lepore, F. (2014). Compensatory plasticity: time matters. Frontiers in human neuroscience, 8, 340. https://doi.org/10.3389/fnhum.2014.00340

Ortiz Alonso, T., Santos, J. M., Ortiz Terán, L., Borrego Hernández, M., Poch Broto, J., & de Erausquin, G. A. (2015). Differences in Early

---

## [Decision Letter · Decision Letter 1]

20 Mar 2024

PONE-D-23-21188R1Early attentional processing and cortical remapping strategies of tactile stimuli in adults with an early and late-onset visual impairment: a cross-sectional studyPLOS ONE

Dear Dr. Sánchez-Sánchez,

Thank you for submitting your manuscript to PLOS ONE. After careful consideration, we feel that it has merit but does not fully meet PLOS ONE’s publication criteria as it currently stands. Therefore, we invite you to submit a revised version of the manuscript that addresses the points raised by reviewer 1.

We look forward to receiving your revised manuscript.

Kind regards,

Valentina Bruno

Academic Editor

PLOS ONE

Journal Requirements:

Reviewers' comments:

Reviewer's Responses to Questions

**Comments to the Author**

1. If the authors have adequately addressed your comments raised in a previous round of review and you feel that this manuscript is now acceptable for publication, you may indicate that here to bypass the “Comments to the Author” section, enter your conflict of interest statement in the “Confidential to Editor” section, and submit your "Accept" recommendation.

Reviewer #1: (No Response)

Reviewer #2: All comments have been addressed

2. Is the manuscript technically sound, and do the data support the conclusions?

Reviewer #1: Partly

Reviewer #2: Partly

3. Has the statistical analysis been performed appropriately and rigorously? 

Reviewer #1: I Don't Know

Reviewer #2: Yes

4. Have the authors made all data underlying the findings in their manuscript fully available?

Reviewer #1: Yes

Reviewer #2: Yes

5. Is the manuscript presented in an intelligible fashion and written in standard English?

Reviewer #1: Yes

Reviewer #2: Yes

6. Review Comments to the Author

Reviewer #1: I acknowledge the work of the authors in answering my previous concerns, and I find the manuscript to be much improved in the revised version. However, I still have 2 comments:

1. Despite the lower type 1 error rate of the Hotelling’s T2 test, results must still be corrected for multiple comparisons for the interaction effect. Perhaps I missed it, but actually I didn't see any information regarding such a correction for the Hotelling's test in the revised version of the manuscript. Hence, I would like to ask the authors if they performed such a correction or not (if not, it has to be added).

2. My second concern regards the fact that the authors did not evaluate the deviant stimuli included in the paradigm. In my opinion since the stimulus was included in the stimulation stream, it had also to be included in the analysis (and it would increase the quality of the research output). I wouldn't like to intervene in the authors' scope, though, so I won't insist on that. Nonetheless, a stronger line of argumentation is needed to justify why it was used in the paradigm, but not in the analysis and interpretation of the data.

Reviewer #2: The manuscript has considerably improved, incorporating the suggestions from all the reviewers.

Though I still consider the explanation of the role of occipital areas as a little over-interpreted, based on the quality of the manuscript, I consider it as acceptable.

7. PLOS authors have the option to publish the peer review history of their article (what does this mean?). If published, this will include your full peer review and any attached files.

Reviewer #1: No

Reviewer #2: No

---

## [Author Response · Author response to Decision Letter 1]

4 May 2024

We thank the editor and the reviewers for their critical review and valuable comments. We have taken into account all of their recommendations and suggestions. Itemized responses are listed below. All the modifications have been clearly highlighted using the "Track Changes" throughout the manuscript to make its revision easier. 

Journal Requirements:

Answer: We have checked the list of references and we have made the following changes: 

- The reference number 59 was a commentary on an original article. Consequently, this reference has been changed to cite the original article. 

- The reference 68 was an erratum for another article. Consequently, this citation has been withdrawn. 

Reviewer #1: I acknowledge the work of the authors in answering my previous concerns, and I find the manuscript to be much improved in the revised version. However, I still have 2 comments:

1. Despite the lower type 1 error rate of the Hotelling’s T2 test, results must still be corrected for multiple comparisons for the interaction effect. Perhaps I missed it, but actually I didn't see any information regarding such a correction for the Hotelling's test in the revised version of the manuscript. Hence, I would like to ask the authors if they performed such a correction or not (if not, it has to be added). 

Answer: We would like to thank the reviewer for his/her valuable comment. In response, we would like to clarify that the results were adjusted for multiple comparisons. This information is now included in the manuscript. Our apologies for any confusion this may have caused. 

Please, see lines 208-211 (manuscript without track changes): For the between groups comparisons, the statistical analyses were conducted in accordance with the comparisons between the three groups, with the probability of type I error set at 0.017. This was calculated by dividing the established error level (0.05) by the number of comparisons (3). 

Please, see line 279 (Table 2): It was considered as statistically significant when p<0.05.

Please, see line 299 (Table 3): It was considered as statistically significant when p<0.017.

2. My second concern regards the fact that the authors did not evaluate the deviant stimuli included in the paradigm. In my opinion since the stimulus was included in the stimulation stream, it had also to be included in the analysis (and it would increase the quality of the research output). I wouldn't like to intervene in the authors' scope, though, so I won't insist on that. Nonetheless, a stronger line of argumentation is needed to justify why it was used in the paradigm, but not in the analysis and interpretation of the data.

Answer: We do thank the reviewer for his/her valuable comment. In this regard, we would like to explain that our objective was to investigate early attentional processing in participants by analysing the P100 wave. This wave is modulated by unconscious and involuntary attention in paradigms where attention must be directed to a particular spatial stimulus (lines 90-92), which justifies the paradigm used in the present study (lines 181-187). In contrast, the analysis of executive responses associated with deviant stimuli is more closely related to cognitive and decision-making processes (Jadhav et al., 2022), which were not the subject of this study. As suggested by the reviewer, this explaining information has been added to the manuscript. 

Please, see lines 423-429 (manuscript without track changes): Furthermore, it should be noted that, as previously stated, the P100 wave is modulated by unconscious and involuntary attention in paradigms where the subject must direct attention to a spatial stimulus [24]. However, the analysis of executive responses associated with deviant stimuli is more closely related to cognitive and decision-making processes [82], which were not the subject of this study and therefore this aspect was not examined. Future studies should focus on the analysis of the deviant stimulus to gain insight into the cognitive and decision-making processes of people with VI.

Reference 

Jadhav, C., Kamble, P., Mundewadi, S., Jaiswal, N., Mali, S., Ranga, S., Suvvari, T. K., & Rukadikar, A. (2022). Clinical applications of EEG as an excellent tool for event related potentials in psychiatric and neurotic disorders. International journal of physiology, pathophysiology and pharmacology, 14(2), 73–83. 

Reviewer #2: 

The manuscript has considerably improved, incorporating the suggestion from all the reviewers. Though I still consider the explanation of the role of occipital areas as a little over-interpreted, based on the quality of the manuscript, I consider it as acceptable. 

Answer: We would like to thank the reviewer for his/her valuable comment. We will take his/her comment into consideration for future manuscripts and thank he/she for considering our manuscript as acceptable for publication.

---

## [Decision Letter · Decision Letter 2]

19 Jun 2024

Early attentional processing and cortical remapping strategies of tactile stimuli in adults with an early and late-onset visual impairment: a cross-sectional study

PONE-D-23-21188R2

Dear Dr. Sánchez-Sánchez,

We’re pleased to inform you that your manuscript has been judged scientifically suitable for publication and will be formally accepted for publication once it meets all outstanding technical requirements.

Kind regards,

Patrick Bruns

Academic Editor

PLOS ONE

Additional Editor Comments (optional):

As you will see below, both original reviewers were happy with your revisions and I agree that the manuscript is now suitable for publication. I noticed two very minor potential inconsistencies in the wording which you could correct in the proof stage if you agree:

Abstract l. 26-27: I believe "...strategies that utilize people with early (EB) and late-onset blindness (LB)..." should read "...strategies that are utilized by people with early (EB) and late-onset blindness (LB)...".

P100 latency section l. 253-254: In this paragraph, add the units of measurement to the mean values given in parentheses (I suppose milliseconds).

Reviewers' comments:

Reviewer's Responses to Questions

**Comments to the Author**

1. If the authors have adequately addressed your comments raised in a previous round of review and you feel that this manuscript is now acceptable for publication, you may indicate that here to bypass the “Comments to the Author” section, enter your conflict of interest statement in the “Confidential to Editor” section, and submit your "Accept" recommendation.

Reviewer #1: All comments have been addressed

Reviewer #2: All comments have been addressed

2. Is the manuscript technically sound, and do the data support the conclusions?

Reviewer #1: Yes

Reviewer #2: Yes

3. Has the statistical analysis been performed appropriately and rigorously? 

Reviewer #1: Yes

Reviewer #2: Yes

4. Have the authors made all data underlying the findings in their manuscript fully available?

Reviewer #1: (No Response)

Reviewer #2: Yes

5. Is the manuscript presented in an intelligible fashion and written in standard English?

Reviewer #1: Yes

Reviewer #2: Yes

6. Review Comments to the Author

Reviewer #1: I would like to thank the authors for carefully answering my concerns. The manuscript is much improved and suitable for publication.

Reviewer #2: (No Response)

7. PLOS authors have the option to publish the peer review history of their article (what does this mean?). If published, this will include your full peer review and any attached files.

Reviewer #1: No

Reviewer #2: **Yes: **S Senthil Kumaran

---

## [Editor Report · Acceptance letter]

29 Jun 2024

PONE-D-23-21188R2 

PLOS ONE

Dear Dr. Sánchez-Sánchez, 

I'm pleased to inform you that your manuscript has been deemed suitable for publication in PLOS ONE. Congratulations! Your manuscript is now being handed over to our production team.

Kind regards, 

on behalf of

Dr. Patrick Bruns 

Academic Editor

PLOS ONE